# Complex Equation Learner: Rational Symbolic Regression with Gradient Descent in Complex Domain

## Abstract

Symbolic regression aims to discover interpretable equations from data, yet modern gradient-based methods fail for operators that introduce singularities or domain constraints, including division, logarithms, and square roots. As a result, Equation Learner-type models typically avoid these operators or impose restrictions, e.g. constraining denominators to prevent poles, which narrows the hypothesis class. We propose a complex weight extension of the Equation Learner that mitigates real-valued optimization pathologies by allowing optimization trajectories to bypass real-axis degeneracies. The proposed approach converges stably even when the target expression has real-domain poles, and it enables unconstrained use of operations such as logarithm and square root. We validate the method on symbolic regression benchmarks and show it can recover singular behavior from experimental frequency response data.

## 1 Introduction

Symbolic regression (SR) and model discovery aim to identify concise, human-interpretable mathematical expressions that govern observed data. In contrast to standard regression methods, which estimate parameters within a fixed functional form, SR jointly searches over both the structure and parameters of candidate models, enabling the direct recovery of explicit analytical relationships from data. This capability is particularly valuable in scientific and engineering domains, such as fundamental and applied physics, cosmology, neuroscience and biology, and complex engineering systems, where interpretable models are essential for extracting physical insight, revealing the underlying mechanisms that govern system behavior, and connecting data-driven discoveries to established theoretical principles. As a result, SR has become a central tool in data-driven scientific discovery, with applications ranging from physics and chemistry to systems biology and materials science.

Symbolic regression methods can be broadly grouped into implicit learned model-based and explicit search-based approaches. Recent years have seen substantial progress in deep learning-based SR, including transformer models Biggio et al. (2021) and reinforcement learning methods Petersen et al. (2021); Tian et al. (2025). However, these approaches remain intrinsically implicit, i.e. they learn an inverse mapping from data to equations tied to the training distribution of expressions, domain, and sampling schemes. Consequently, their behavior under distribution shift cannot be systematically controlled especially in extrapolative regimes. In contrast, explicit search-based methods do not rely on such learned priors and are typically not sensitive to the expression structure changes compared to the training setup. Among these, genetic programming (GP) has historically been a dominant paradigm for symbolic regression Makke & Chawla (2024). However, GP remains inherently heuristic and combinatorial, with performance depending strongly on the search settings, operator set, and problem structure.

Another class of SR methods formulates model discovery as a gradient-based optimization problem by minimizing data approximation error Makke & Chawla (2024). Some approaches reduce SR to linear regression over a fixed library of functions, such as Sparce Identification of Nonlinear Dynamics (SINDy) Brunton et al. (2016), which selects a sparse linear combination of of candidate library terms. SINDy performs well when true dynamics are sparse in the chosen basis, but degrades when essential functional forms are absent from

the library and becomes less scalable as the library growths Brunton et al. (2016). More expressive non-linear models, such as Kolmogorov-Arnold Networks (KANs), learn univariate edge functions that can yield interpretable components but require post-processing to obtain compact closed-form equations Liu et al. (2025). The Equation Learner (EQL) instead builds hypotheses directly from analytic operators in a differentiable architecture, enabling extraction of explicit symbolic expressions after training Martius & Lampert (2016). Extensions incorporating modified division operators increase EQL's expressivity Sahoo et al. (2018), yet they enforce a fixed sign on the denominator, substantially restricting the class of representable rational forms.

Among gradient-based approaches, the EQL is seen as the most promising approach, as its architecture explicitly composes operators within a differentiable network. By construction, this model contains symbolic expressions generated by the chosen operator library up to a predefined compositional depth, while the learning procedure reduces to optimizing internal weights. However, in its current form, EQL is limited to a relatively narrow set of operations. Fundamental operators such as division, logarithms, and square root are difficult to incorporate reliably due to intrinsic optimization pathologies, as analyzed in Section 2. These operators are ubiquitous in physical laws and constitutive models, and their exclusion substantially restricts the scope of symbolic model discovery. This limitation motivates the need for a principled symbolic regression framework that preserves explicit, interpretable structure while supporting stable optimization over a substantially broader class of nonlinear operators, including those that may induce singularities.

To address these gaps, we propose the Complex Equation Learner (CEQL), an extension of EQL that enables stable gradient-based learning of rational expressions involving division. CEQL relaxes real-valued training by allowing network weights to evolve in the complex domain while projecting model outputs back onto the real axis. Optimization in the complex space allows training trajectories to bypass real-axis degeneracies that, in purely real-valued optimization, lead to sign-changing gradient cancellations. As a result, CEQL enables the reliable learning of explicit denominators and near-singular structures using gradient descent. We demonstrate that CEQL identifies poles in rational target expressions and recovers interpretable pole-bearing symbolic forms. In particular, CEQL discovers rational models that capture resonant behavior in frequency response functions, where inspection of the learned denominator directly relates pole locations to changes in system parameters. Moreover, operating in the complex space removes the standard real-valued domain restrictions of operators such as logarithms and square roots, allowing them to be incorporated into a differentiable symbolic architecture without ad hoc constraints. An ablation study further isolates the effect of the complex-valued parametrization, the imaginary-weight penalty, skip connections, and operator-library choice on multivariate rational recovery.

## 2 Preliminaries

### 2.1 Gradient Pathologies in Real-Valued Optimization

Consider the problem of approximating the function $f(x) = 1/x$ on the interval $[-\ell, \ell]$. Let $\{x_i\}_{i=1}^N$ be a set of sample points drawn from this interval, and let the parametric model be $\hat{f}(x) = 1/(x + a)$, where $a \in \mathbb{R}$. We formulate the approximation task as the optimization problem

$$a^\star = \arg\min_{a \in \mathbb{R}} \mathcal{L}(a), \quad \mathcal{L}(a) = \frac{1}{N} \sum_{i=1}^N \left( f(x_i) - \hat{f}(x_i) \right)^2.$$

Here, $a^\star$ denotes the minimizer of the loss $\mathcal{L}(a)$, which in this setting is attained at $a^\star = 0$. Despite its apparent simplicity, this problem exhibits fundamental obstacles for gradient-based optimization methods. To see this, consider the gradient of $\mathcal{L}$ with respect to $a$:

$$\begin{aligned}
\frac{\partial \mathcal{L}}{\partial a} &= \frac{\partial}{\partial a} \left( \frac{1}{N} \sum_{i=1}^N \left( \frac{1}{x_i} - \frac{1}{x_i + a} \right)^2 \right) \\
&= \frac{2}{N} \sum_{i=1}^N \left( \frac{1}{x_i} - \frac{1}{x_i + a} \right) \frac{1}{(x_i + a)^2}.
\end{aligned} \tag{1}$$

Let

$$g(x_i, a) = \left( \frac{1}{x_i} - \frac{1}{x_i + a} \right) \frac{1}{(x_i + a)^2}$$

denote the per-sample contribution to the gradient. The behavior of the total gradient is governed by this term, which simplifies to $g(x_i, a) = a \left( x_i(x_i + a)^3 \right)^{-1}$. For any fixed $a$, the sign of $g(x_i, a)$ is therefore determined by the signs of $x_i$ and $x_i + a$. Consequently, the per-sample gradient contribution $g(x_i, a)$ changes sign at $x_i = 0$ and $x_i = -a$, producing opposite gradient contributions from different intervals of the domain. When samples lie on both sides of these sign-change boundaries, the terms $g(x_i, a)$ in the gradient sum cancel, leading to oscillatory or near-zero gradients. As a result, the total gradient does not, in general, provide a consistent descent direction toward the minimizer $a^\star = 0$. The same sign-changing mechanism arises whenever denominators introduce additional terms, as each term induces further sign-changes in the gradient.

This observation shows that the difficulty encountered above is not a consequence of poles in the model, nor of isolated pathological points in the domain. It originates from the intrinsic sign-changing structure of the gradient in the real-valued parameter space, which leads to substantial cancellations across the data and results in a flat loss landscape. An analogous effect arises for the gradient when approximating $f = \sin(x)$ with the parametric model $\hat{f}(x) = \sin(bx)$, one can derive the gradient of the loss w.r.t. parameter $b$, and verify that the per-sample gradient contributions similarly change sign across the domain, leading to the similar cancellation mechanism. Thus, the main challenge faced by gradient-based methods is a structural property of the function's behavior in the real-domain, rather than an artifact of poles.

## 2.2 Multi-valued Functions and Domain Restrictions

In addition to the gradient pathologies discussed above, certain operators introduce a distinct class of difficulties related to domain restrictions induced by the branch cuts of multi-valued functions. A representative example is the logarithm. In the real domain, the natural logarithm $\ln(\cdot)$ is defined only for strictly positive arguments. When implemented within a real-valued EQL network, the input to the logarithm corresponds to an intermediate activation produced by preceding layers and therefore varies across samples during training. As a consequence, some samples may drive this activation outside the domain of definition, leading to undefined forward evaluations whenever the activation is negative or approaches zero.

From a complex-analytic perspective, this restricted real-domain definition reflects the fact that the logarithm is multi-valued in the complex plane. For a complex argument $z \in \mathbb{C} \setminus \{0\}$, the logarithm admits the family of values

$$\ln z = \ln |z| + i(\arg z + 2\pi k), \qquad k \in \mathbb{Z}.$$

Any single-valued realization of $\ln(\cdot)$ therefore requires the selection of a branch, which necessarily introduces a branch cut along which the argument $\arg z$ is discontinuous. The domain restriction of the real logarithm can thus be interpreted as a consequence of this branch selection, rather than as an intrinsic property of the logarithm itself.

The same considerations apply to the square root operation. The square root $\sqrt{x}$ is commonly expressed using the identity

$$\sqrt{x} = x^{\frac{1}{2}} = \exp\left( \frac{1}{2} \ln x \right),$$

with $x \in \mathbb{R}$. This representation makes explicit that expressions involving $\sqrt{x}$ inherit the same branch-cut-related issues as the logarithm. In particular, unless the base $x$ is constrained to remain positive, the expression becomes ill-defined in the real domain.

A practical workaround, similar in spirit to sign-restricted division used in previous EQL extensions Sahoo et al. (2018), is to assume that the argument of the logarithm does not change sign and to replace $\ln(x)$ with $\ln |x|$, and correspondingly $\sqrt{x}$ with $\exp\left( \frac{1}{2} \ln |x| \right)$. While this modification helps to avoid undefined regions in the real-valued formulation, it imposes a strong structural assumption. In realistic settings, the true target expression may change sign, with the logarithm being well-defined only on subsets of the domain actually covered by the data. Enforcing a globally non-sign-changing behavior in the symbolic network therefore

prevents the recovery of such expressions, even when all observed samples lie within regions where the target expression itself is well-defined.

## 3 Methods

### 3.1 Surrogate Symbolic Operations

To address the challenges described in Section 2, we propose to modify the backbone of the EQL framework by allowing its parameters to take complex values. This complex parametrization enlarges the optimization domain from $\mathbb{R}^p$ to $\mathbb{C}^p$, thereby altering the geometry of the loss landscape and mitigating gradient pathologies that arise in purely real-valued optimization. Model outputs are subsequently projected onto the real axis, and the loss is computed using the real part of the network output. Throughout this section, for any complex activation $z = a + ib$ with $a, b \in \mathbb{R}$, we denote by $\Re(z) = a$ its real part. For vector-valued or batch-valued activations, $\Re(\cdot)$ is applied element-wise.

**For division operation**, complex-valued parameters introduce a complex shift in the denominator. In the example of approximating $f(x) = 1/x$ with a model of the form $\hat{f}(x) = 1/(x + a)$, allowing $a \in \mathbb{C}$ smooths the optimization trajectory by avoiding gradient cancellation effects present in real-valued optimization. An illustrative optimization example is given in Appendix A.

Gradient cancellations arise primarily for parameters appearing in denominators and do not occur for parameters in numerators. Accordingly, we define a surrogate division operation $b_{\mathrm{div}}$ as

$$b_{\mathrm{div}}(x, y) = \Re(\Re(x)/y), \tag{2}$$

where $x, y \in \mathbb{C}$ are the inputs to the surrogate operator. This design preserves a real-valued symbolic output while retaining extra degree of freedom in the denominator, thereby mitigating the gradient-cancellation effects encountered in real-valued optimization.

**The logarithm and square root** are implemented as real-projected complex surrogates. For a complex activation $z \in \mathbb{C}$, we define

$$u_{\log}^*(z) = \Re(\log z) + 0i, \tag{3}$$

where $\log$ is the principal complex logarithm. Similarly,

$$u_{\sqrt{\cdot}}^*(z) = \Re(\sqrt{z}) + 0i, \tag{4}$$

where $\sqrt{z}$ is the principal complex square root.

This construction is used only as an optimization surrogate. All inputs, targets, and final predictions in this work are real-valued. On the positive real axis, the surrogates coincide with the standard real operators, i.e. $u_{\log}(x) = \log x$ and $u_{\sqrt{\cdot}}(x) = \sqrt{x}$ for $x > 0$. For intermediate activations outside the positive real axis, the complex evaluation avoids undefined forward passes during training. This is relevant for expressions such as E-5 and E-6, where the logarithm and square-root arguments are quadratic functions that change sign over the nominal training interval. Samples are used only where the corresponding real-valued target is defined, but the learned intermediate activations are not constrained to remain positive during optimization. Unlike replacements such as $\log|\cdot|$ or $\sqrt{|\cdot|}$, the proposed surrogates do not change the real-valued target expression on its domain of definition.

All other unary operators $u_i$ are implemented via real-projected surrogates $u_i^*(x)$ defined as

$$u_i^*(x) = u_i(\Re(x)) + 0i, \tag{5}$$

where $x \in \mathbb{C}$ denotes the complex-valued input activation. That is, the original real-valued operator is applied to the real part of the activation, and the result is embedded back into $\mathbb{C}$ with zero imaginary component.

Similarly, for binary operators $b_i$, i.e., operators with two arguments, we define the surrogate $b_i^*$ as

$$b_j^*(x, y) = b_j(\Re(x), \Re(y)) + 0i, \tag{6}$$

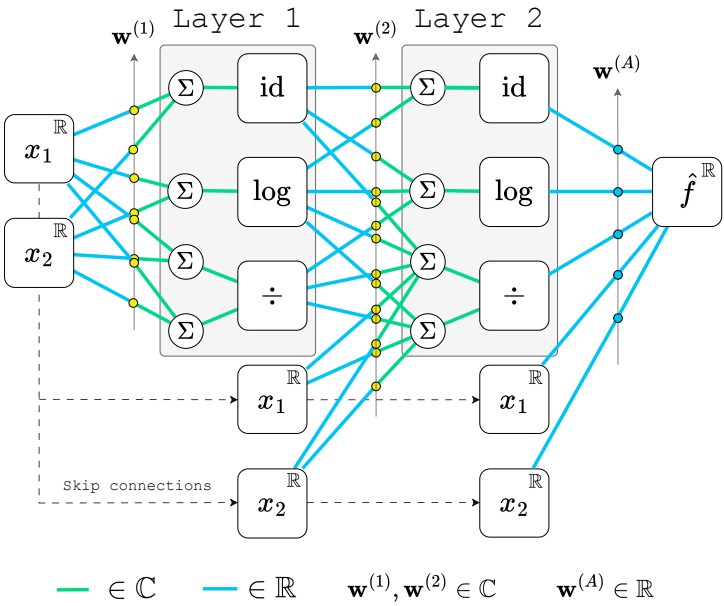

Figure 1: Complex Equation Learner (CEQL) architecture. Internal weights are complex-valued, enabling optimization to bypass real-valued degeneracies induced by division and multivaled operators. While the internal weights are optimized in $\mathbb{C}$, the output of CEQL is projected to $\mathbb{R}$ to minimize discrepancy with the real-valued target values.

with $x, y \in \mathbb{C}$. This construction is used for all binary operators except division, which is handled by the dedicated surrogate $b_{\mathrm{div}}$ defined above.

Overall, these surrogate definitions enforce a real-valued symbolic hypothesis class at the level of operator evaluations, while still allowing complex-valued parameters and intermediate representations to introduce additional degrees of freedom during optimization for selected operators.

### 3.2 Complex Equation Learner

The proposed CEQL approach builds on the original EQL framework. As a fully connected neural network, CEQL can be viewed as a directed acyclic graph, in which nodes implement symbolic operations and edges carry learnable parameters. In contrast to EQL, CEQL uses complex-valued weights, which enlarge the optimization space and address the division-related gradient issues discussed in Section 2.1. The graph contains one input node per input variable of the target function and a single output node representing the CEQL prediction, as illustrated in Figure 1. While internal signals within CEQL are complex, both the inputs and the target outputs are strictly real-valued in all experiments.

Each layer contains a fixed set of operators, referred to as the operator library. In this work, the library consists of unary surrogates $u_j^*$ and binary surrogates $b_k^*$ defined in Section 3.1. The inputs to the operators in layer $L$ are formed by summation nodes,

$$z_j^{(L)} = \sum_{i=1}^{d_{L-1}} w_{ij}^{(L)} h_i^{(L-1)}, \qquad j = 1, \ldots, m + 2n,$$

where $h_i^{(L-1)}$ denotes the $i$-th activation of layer $L-1$, $d_{L-1}$ is the number of activations in that layer, and $w_{ij}^{(L)}$ is the (complex-valued) weight on the edge from activation $i$ to summation node $j$. The first $m$ summation nodes are passed through unary operators, $h_j^{(L)} = u_j^*(z_j^{(L)})$ for $j = 1, \ldots, m$. The remaining

$2n$ summation nodes are grouped into $n$ consecutive pairs and passed to the binary operators, $h_{m+k}^{(L)} = b_k^*(z_{m+2k-1}^{(L)}, z_{m+2k}^{(L)})$ for $k = 1, \ldots, n$. The model output $\hat{f}$ is obtained as a weighted sum of the final-layer activations.

Beyond the original EQL formulation, CEQL incorporates skip connections that route the original input variables directly to deeper layers by concatenating them with intermediate activations. This design is inspired by residual and feature-concatenation connectivity patterns in deep networks He et al. (2016). Skip connections improve gradient propagation when the target expression is shallow and allow the optimizer to select shorter computational paths, thereby simplifying the search over expression structures.

Overall, the resulting architecture spans the space of symbolic expression trees generated by the chosen library up to a prescribed depth. CEQL is trained to minimize the discrepancy between the predicted output and the target values while enforcing sparsity. The sparsity mechanism and optimization procedure are described in Section 3.3.

## 3.3 Optimization Strategy

CEQL is trained by minimizing mean squared error (MSE) between the target values and the real part of the CEQL output, while internal parameters remain complex-valued. The $\ell_2$ objective is augmented with auxiliary terms that promote sparsity and stabilize optimization. Training is organized into three phases.

In the first phase, we minimize the training MSE augmented with a sparsity-inducing penalty on the weights and a penalty on their imaginary components. The sparsity term encourages small-magnitude connections to shrink toward zero, while the imaginary penalty biases parameters toward the real axis, keeping intermediate evaluations of multi-valued operators close to the principal branch. The corresponding regularization coefficients are chosen several orders of magnitude smaller than the MSE term. At the end of this phase, all connections with magnitude below a fixed threshold are pruned and subsequently held at zero. We then perform a cascade cleanup to remove edges and nodes that are no longer connected to the output.

In the second phase, training continues on the pruned model using a stronger sparsity penalty and the same imaginary-weight penalty. To further simplify the expression and accelerate convergence, we perform iterative pruning cycles during training. At each cycle, a prescribed fraction of connections with the smallest impact on the batch loss is removed, while enforcing a minimum number of remaining edges to control the maximum size of the resulting expression tree. After each pruning step, disconnected branches are again removed via cascade cleanup.

In the third phase, sparsity regularization is disabled and the remaining parameters are fine-tuned using only the MSE loss and the small imaginary-weight penalty. This final refinement improves numerical accuracy while preserving the discovered symbolic structure. The final symbolic expression is obtained by extracting the expression tree induced by the remaining active connections in the CEQL architecture.

## 3.4 Evaluation of Discovered Symbolic Models

Symbolic regression aims to fit observed data with an explicit formula that is both accurate and compact. Many existing works report symbolic criteria such as exact match or algebraic equivalence. In practice, however, these criteria are brittle: algebraic simplification is not always reliable, and many distinct expressions can fit the data equally well while remaining non-equivalent or difficult to compare symbolically. This concern is consistent with recent benchmark studies showing that standard SR datasets and evaluation protocols can overestimate scientific-discovery performance when they rely only on simplified sampling regimes or coarse symbolic metrics Matsubara et al. (2024).

In this work, we primarily evaluate models using mean squared error (MSE) on both an interpolation test set (within the training domain) and an extrapolation test set (outside the training domain). While interpolation assesses how well a model fits data within the observed range, extrapolation evaluates its ability to generalize beyond it. Specifically, the extrapolation set is constructed by sampling inputs from regions that lie strictly outside the training domain, ensuring that the model is evaluated on previously unseen value ranges.

Table 1: The set of SR benchmark expressions.

| # | Expression | Ill-posedness | # | Expression | Ill-posedness |
|---|---|---|---|---|---|
| E-1 | $1.87\,x_1 + 2.01$ | – | E-6 | $2.31\,\sqrt{2.52\,x_1^2 - 1.52\,x_1 - 2.24}$ | undef. |
| E-2 | $1.56\,x_1 + 1.59\,x_2 - 2.91$ | – | E-7 | $\dfrac{0.53 - 2.94\,x_1}{2.32\,x_1 + 1.80}$ | pole (1) |
| E-3 | $2.48\,x_1^2 + 1.92\,x_1 - 0.68$ | – | E-8 | $\dfrac{1.00\,x_1 + 2.48\,x_2 - 1.36}{2.26\,x_1 - 0.91\,x_2 + 1.94}$ | pole (1) |
| E-4 | $0.55\,x_1^2 + 2.45\,x_1 x_2 + 2.95\,x_2^2$ $+1.65\,x_1 + 0.80\,x_2 + 0.86$ | – | E-9 | $\dfrac{2.84\,x_1^2 + 1.84\,x_1 - 2.33}{-0.66\,x_1^2 + 2.94\,x_1 + 1.35}$ | pole (1) |
| E-5 | $-2.05\,\log(1.56\,x_1^2 - 0.55\,x_1 - 2.15)$ | undef. | E-10 | $\dfrac{-1.08\,x_1^2 - 2.85\,x_1 - 2.08}{2.56\,x_1^2 + 1.78\,x_1 - 0.74}$ | pole (2) |

This distinction is particularly important for symbolic regression, whose objective is to recover the underlying functional relationship rather than to interpolate between samples. In the synthetic benchmarks considered here, a compact ground-truth expression is known to exist by construction. Models that merely rely on flexible function approximation can achieve low interpolation error but typically fail to generalize outside the training domain. In contrast, models that correctly identify the underlying functional form are expected to maintain low error in extrapolation. Consequently, low extrapolation error serves as a practical proxy for recovering the true governing expression up to algebraic equivalence, and provides a more stringent measure of robustness.

To assess the structural properties of the recovered expressions, we additionally report the node count (NC) of the inferred expression tree and the symbolic recovery rate (RR). RR is defined as the fraction of independent runs in which the recovered expression is symbolically equivalent to the ground-truth expression after coefficient rounding and symbolic simplification.

## 4 Results

This section evaluates the performance of the proposed CEQL method against representative symbolic regression baselines from three major methodological families: the GP-based PySR algorithm, the linear sparse regression method SINDy, and the non-linear EQL variant with division. We then perform an ablation study to isolate the effects of complex-valued weights, the imaginary-weight penalty, skip connections, and operator-library choice. Finally, we demonstrate the capabilities on a real-world task by approximating the frequency response function of a cantilever steel beam.

### 4.1 Symbolic Regression Benchmarks

To evaluate performance of the proposed method across increasing structural complexity, presence of singularities or domain restrictions, we construct a controlled benchmark set of analytic expressions. The benchmark expressions, summarized in Table 1, range from simple linear and polynomial forms to expressions involving rational functions, logarithms, and square roots.

The expressions are generated symbolically and evaluated exactly. Coefficients are sampled independently from a signed uniform distribution $c = s \cdot u$, with $s \in \{-1, +1\}$ and $u \sim \mathcal{U}(0.5, 3.0)$, and rounded to two decimal places. For each expression, three separate datasets are generated: a training set, an interpolation test set sampled from the same domain, and an extrapolation test set sampled strictly outside the training domain.

We evaluate symbolic models discovered by CEQL in both interpolation and extrapolation regimes. Training and interpolation samples are drawn from $[-2, 2]^d$, whereas extrapolation samples are drawn from the disjoint outer region $([-4, -2] \cup [2, 4])^d$, ensuring evaluation on previously unseen regions of the input space. The datasets contain 128 training points and 8192 points for both interpolation and extrapolation testing. Samples producing non-finite, undefined, or values exceeding $10^2$ are discarded. This evaluation setup is

particularly relevant for assessing SR models under assumption that the underlying governing model has a compact symbolic form. SR models that identify the correct functional form are expected to generalize consistently across domains, including regions outside the training range.

Beyond polynomials, the benchmark expressions are constructed to expose structural challenges for gradient-based symbolic regression methods. Rational expressions contain denominators with one or more poles inside the training domain, while explicitly excluding poles from the extrapolation domain. Logarithmic and square-root expressions use quadratic arguments that change sign within the training domain, creating intervals where the target function is undefined. In real-valued formulations, such structures induce ill-conditioned loss landscapes, and invalid forward evaluations.

Table 2: Experimental results of SR methods applied to expressions with singularities and domain constraints. The table reports the mean value of the metric and its standard deviation obtained across 5 independent runs. For deterministic SINDy, evaluations are reported from the single run.

| # | PySR Interp. MSE | PySR Extrap. MSE | SINDy Interp. MSE | SINDy Extrap. MSE |
|---|---|---|---|---|
| E-1 | $(1.6 \pm 2.4) \times 10^{-15}$ | $(2.4 \pm 3.3) \times 10^{-15}$ | $0.0$ | $0.0$ |
| E-2 | $(6.0 \pm 7.3) \times 10^{-15}$ | $(1.7 \pm 2.1) \times 10^{-14}$ | $5.1 \times 10^{-30}$ | $1.5 \times 10^{-29}$ |
| E-3 | $(4.6 \pm 4.7) \times 10^{-15}$ | $(2.7 \pm 3.2) \times 10^{-14}$ | $1.5 \times 10^{-30}$ | $6.9 \times 10^{-29}$ |
| E-4 | $(5.8 \pm 5.5) \times 10^{-9}$ | $(2.4 \pm 2.7) \times 10^{-7}$ | $5.7 \times 10^{-29}$ | $4.9 \times 10^{-28}$ |
| E-5 | $(6.4 \pm 10.3) \times 10^{-5}$ | $(3.6 \pm 3.2) \times 10^{-7}$ | $1.1$ | $1.7 \times 10^{3}$ |
| E-6 | $(9.1 \pm 11.9) \times 10^{-14}$ | $(1.0 \pm 0.8) \times 10^{-13}$ | $1.3 \times 10^{-1}$ | $1.3 \times 10^{2}$ |
| E-7 | $(4.1 \pm 8.1) \times 10^{-5}$ | $(1.7 \pm 3.1) \times 10^{-12}$ | $5.0 \times 10^{1}$ | $6.5 \times 10^{0}$ |
| E-8 | $(2.9 \pm 2.9) \times 10^{-4}$ | $(2.5 \pm 3.5) \times 10^{-4}$ | $7.1 \times 10^{1}$ | $2.1 \times 10^{2}$ |
| E-9 | $(5.1 \pm 6.6) \times 10^{-5}$ | $(2.1 \pm 2.6) \times 10^{0}$ | $3.1 \times 10^{1}$ | $3.3 \times 10^{1}$ |
| E-10 | $(8.0 \pm 10.3) \times 10^{-8}$ | $(6.4 \pm 6.7) \times 10^{-13}$ | $4.6 \times 10^{1}$ | $4.0 \times 10^{1}$ |

| # | CEQL (our) Interp. MSE | CEQL (our) Extrap. MSE | EQL$_\div$ Interp. MSE | EQL$_\div$ Extrap. MSE |
|---|---|---|---|---|
| E-1 | $(1.0 \pm 0.8) \times 10^{-12}$ | $(3.6 \pm 4.5) \times 10^{-12}$ | $(8.2 \pm 6.2) \times 10^{-7}$ | $(5.3 \pm 8.0) \times 10^{-3}$ |
| E-2 | $(2.7 \pm 1.5) \times 10^{-12}$ | $(7.5 \pm 3.9) \times 10^{-12}$ | $(5.7 \pm 6.4) \times 10^{-3}$ | $(2.9 \pm 5.4) \times 10^{-1}$ |
| E-3 | $(6.8 \pm 10.1) \times 10^{-11}$ | $(2.0 \pm 3.2) \times 10^{-9}$ | $(1.1 \pm 0.7) \times 10^{-1}$ | $(1.3 \pm 0.7) \times 10^{2}$ |
| E-4 | $(9.0 \pm 5.8) \times 10^{-11}$ | $(2.4 \pm 2.1) \times 10^{-9}$ | $(6.2 \pm 3.0) \times 10^{-1}$ | $(4.2 \pm 5.9) \times 10^{5}$ |
| E-5 | $(5.9 \pm 7.2) \times 10^{-9}$ | $(3.4 \pm 4.3) \times 10^{-11}$ | $(1.9 \pm 1.1) \times 10^{0}$ | $(7.5 \pm 14.9) \times 10^{2}$ |
| E-6 | $(3.8 \pm 4.7) \times 10^{-7}$ | $(2.1 \pm 3.7) \times 10^{-9}$ | $(4.7 \pm 9.1) \times 10^{-1}$ | $(3.6 \pm 3.8) \times 10^{1}$ |
| E-7 | $(8.6 \pm 7.7) \times 10^{-8}$ | $(1.9 \pm 3.7) \times 10^{-9}$ | $(1.3 \pm 2.3) \times 10^{3}$ | $(1.8 \pm 3.6) \times 10^{3}$ |
| E-8 | $(9.3 \pm 14.2) \times 10^{-8}$ | $(1.5 \pm 2.4) \times 10^{-9}$ | $(1.3 \pm 2.2) \times 10^{3}$ | $(7.1 \pm 14.0) \times 10^{4}$ |
| E-9 | $(4.8 \pm 3.5) \times 10^{-8}$ | $(9.7 \pm 19.4) \times 10^{-7}$ | $(1.3 \pm 1.0) \times 10^{2}$ | $(8.4 \pm 8.5) \times 10^{1}$ |
| E-10 | $(2.7 \pm 1.1) \times 10^{-6}$ | $(4.2 \pm 6.7) \times 10^{-11}$ | $(7.0 \pm 9.2) \times 10^{2}$ | $(7.9 \pm 13.4) \times 10^{0}$ |

Table 2 summarizes the experimental results of different SR methods applied to the generated benchmark datasets. We compare the proposed CEQL method against SINDy, EQL$_\div$, and PySR, a state-of-the-art GP-based SR algorithm. The table reports the mean squared error (MSE) on the interpolation and extrapolation test sets across five independent runs. Structural complexity and symbolic recovery are reported separately in Table 3. The hyperparameter settings for all methods are reported in Appendix B.

As expected for a fixed-library sparse regression method, SINDy achieves extremely low test errors on polynomial expressions, where the target functions are sparse in the chosen polynomial library. However, it does not recover the nonlinear benchmarks E-5–E-10, including the rational, logarithmic, and square-root expressions. This behavior is largely expected, since the required features would need to appear explicitly in the SINDy library. In particular, SINDy does not learn the internal coefficients of nonlinear arguments or denominators, representing all possible rational terms, logarithmic arguments, and square-root arguments through a fixed library would require a prohibitively large set of candidate terms. Thus, these results mainly illustrate the limitation of fixed-library linear SR when the correct nonlinear features are absent.

Table 3: Structural complexity and symbolic recovery rate on symbolic regression benchmarks. NC denotes the node count of the recovered symbolic graph. RR denotes the symbolic recovery rate, i.e. the fraction of independent runs in which the recovered expression is symbolically equivalent to the ground-truth expression.

| # | CEQL (our) | | PySR | | SINDy | | EQL$_{\div}$ | |
| --- | --- | --- | --- | --- | --- | --- | --- | --- |
| | NC | RR | NC | RR | NC | RR | NC | RR |
| E-1 | $5.0 \pm 0.0$ | 1.00 | $6.8 \pm 3.6$ | 1.00 | 5.0 | 1.00 | $37.0 \pm 5.8$ | 0.00 |
| E-2 | $8.0 \pm 0.0$ | 1.00 | $8.0 \pm 0.0$ | 1.00 | 8.0 | 1.00 | $115.4 \pm 38.1$ | 0.00 |
| E-3 | $10.0 \pm 0.0$ | 1.00 | $13.0 \pm 1.1$ | 1.00 | 10.0 | 1.00 | $43.0 \pm 0.0$ | 0.00 |
| E-4 | $22.0 \pm 0.0$ | 1.00 | $19.6 \pm 0.5$ | 1.00 | 22.0 | 1.00 | $151.0 \pm 0.0$ | 0.00 |
| E-5 | $18.4 \pm 4.3$ | 0.80 | $20.2 \pm 5.7$ | 0.40 | 19.0 | 0.00 | $43.0 \pm 0.0$ | 0.00 |
| E-6 | $21.4 \pm 9.3$ | 1.00 | $15.0 \pm 3.3$ | 0.80 | 19.0 | 0.00 | $43.0 \pm 0.0$ | 0.00 |
| E-7 | $26.8 \pm 12.4$ | 0.80 | $15.0 \pm 3.3$ | 0.80 | 19.0 | 0.00 | $43.0 \pm 0.0$ | 0.00 |
| E-8 | $33.2 \pm 8.4$ | 1.00 | $16.8 \pm 1.0$ | 1.00 | 46.0 | 0.00 | $151.0 \pm 0.0$ | 0.00 |
| E-9 | $37.2 \pm 9.6$ | 1.00 | $20.6 \pm 1.0$ | 0.60 | 19.0 | 0.00 | $43.0 \pm 0.0$ | 0.00 |
| E-10 | $29.4 \pm 3.2$ | 1.00 | $18.6 \pm 2.2$ | 1.00 | 19.0 | 0.00 | $43.0 \pm 0.0$ | 0.00 |

PySR and CEQL approaches achieve in general low errors on the benchmark expressions. Overall, CEQL achieves performance comparable to the strong GP-based PySR baseline, while substantially improving over EQL-type gradient-based models on expressions with poles and domain restrictions, demonstrating that the proposed complex-domain formulation alleviates the key optimization difficulties of EQL. As shown in Table 3, for simpler expressions, such as E-1–E-4 and E-6, the zero variance in the node count indicates that CEQL consistently converges to the same sparsified symbolic structure. For more complex rational expressions, the non-zero variance in NC suggests that additional terms with negligible impact on MSE may persist in some runs. This behavior is likely due to the larger range of the target values in rational expressions, which reduces the sensitivity of the loss to small residual terms. In addition to the error-based metrics, Table 3 reports symbolic recovery rates. PySR and CEQL show comparably high recovery rates on the benchmark set, which is consistent with their low interpolation and extrapolation errors. In contrast, the very low recovery rate of EQL$_{\div}$ is connected to the difficulty of eliminating redundant terms without an effective pruning procedure. The recovered symbolic expressions with the lowest training MSE across five independent CEQL runs are provided in Appendix C.

## 4.2   Ablation study

To assess the role of the main CEQL components, we perform an ablation study on a randomly generated three-variate rational expression,

$$f(x_1, x_2, x_3) = \frac{-2.09x_1 + 1.17x_2 + 0.60x_3 + 0.54}{-2.02x_1 + 2.32x_2 + 1.86x_3 - 2.84}. \tag{7}$$

The target contains an affine denominator depending on all three variables, and therefore requires recovery of a sign-changing denominator hyperplane. The expression provides a controlled test of multivariate rational recovery, where the learned model must identify both the numerator and the denominator structure from data.

We compare the full CEQL model with four variants: CEQL without the imaginary-weight penalty, CEQL without skip connections, CEQL with additional logarithmic operators in the library, and a real-valued EQL variant. The logarithmic library variant is included because, in preliminary experiments, we observed that unnecessary logarithmic operators can disturb rational discovery. A possible explanation is that logarithmic and rational nodes compete during optimization, causing their coefficients to be underestimated and increasing the risk that required rational terms are removed during pruning.

As expected, the real-valued variant does not recover the target expression in any run and gives the largest extrapolation error. This is consistent with the intuition in Section 2.1: for rational expressions with sign-changing denominators, purely real-valued optimization can suffer from gradient cancellation and unstable descent directions. The fact that the complex-valued variants recover the expression in several runs provides empirical support for introducing the additional imaginary dimension.

Table 4: Ablation study on the randomly generated three-variate rational expression in Equation 7. The table reports interpolation MSE, extrapolation MSE, modal extrapolation MSE, node count (NC), and symbolic recovery rate (RR) across five independent runs.

| Model variant | Interp. MSE | Extrap. MSE | Modal extrap. MSE | NC | RR |
|---|---|---|---|---|---|
| CEQL | $(6.99 \pm 15.7) \times 10^0$ | $(3.46 \pm 7.7) \times 10^0$ | $2.54 \times 10^{-7}$ | $23.0 \pm 9.7$ | 0.80 |
| CEQL without imaginary penalty | $(2.87 \pm 6.4) \times 10^0$ | $(3.39 \pm 7.6) \times 10^0$ | $1.96 \times 10^{-4}$ | $29.6 \pm 8.0$ | 0.00 |
| CEQL without skip connections | $(1.82 \pm 4.1) \times 10^1$ | $(3.59 \pm 8.0) \times 10^1$ | $5.56 \times 10^{-9}$ | $40.4 \pm 12.0$ | 0.80 |
| CEQL with log | $(2.32 \pm 1.9) \times 10^1$ | $(1.21 \pm 0.88) \times 10^1$ | $5.43 \times 10^0$ | $56.8 \pm 52.0$ | 0.20 |
| Real-valued EQL | $35.36 \pm 0.05$ | $(3.60 \pm 3.1) \times 10^1$ | $2.26 \times 10^1$ | $23.2 \pm 26.0$ | 0.00 |

The full CEQL model recovers the correct symbolic expression in four out of five runs. Its low modal extrapolation error shows that, when the recovered structure is consistent across runs, the learned expression generalizes accurately outside the training domain. Removing skip connections preserves the same recovery rate, but increases the average errors and the node count, suggesting that skip connections are not required for representability here, but help stabilize the search for compact structures.

Removing the imaginary-weight penalty gives low average numerical errors but zero symbolic recovery. This indicates that accurate prediction alone does not guarantee recovery of the symbolic form. Without this penalty, the model can exploit complex-valued degrees of freedom to fit the data while ending in expressions that are not algebraically equivalent to the real-valued target.

Adding logarithmic operators decreases the recovery rate from 0.80 to 0.20 and substantially increases the node-count variance. Since the target is purely rational, these operators do not add useful representational capacity. Instead, they enlarge the search space and may introduce competing branches during optimization, which is consistent with the practical observation that unnecessary logarithmic terms can interfere with rational recovery and pruning. Overall, the ablation results indicate that complex-valued weights are essential for learning sign-changing denominators, while the imaginary-weight penalty and skip connections improve the stability of symbolic recovery.

### 4.3 Frequency Response Function Discovery

This experiment is designed to evaluate if CEQL can recover pole-like structure from real measured data, rather than only from synthetic benchmarks. This experiment complements the controlled synthetic benchmarks by testing CEQL on measured frequency response data, with the focus on interpretable pole-like structure and damage-dependent resonance shifts rather than exact recovery of a known closed-form expression. Frequency response functions of lightly damped structures exhibit sharp resonance peaks whose locations shift with changes in attached masses; these peaks are naturally explained by rational forms with denominators that approach zero near modal frequencies. The goal is therefore to learn an interpretable symbolic surrogate of the measured response magnitude that can represent both the broadband trend and the resonance spikes, and whose denominators can be inspected to understand how resonance frequencies shift as a function of structural damage.

The dataset consists of repeated experimental measurements of the frequency response of a cantilever steel beam equipped with six detachable masses de Sousa & Machado (2024). Each experiment yields a complex-valued inertance response over $f \in [0, 2000]$ Hz, recorded as magnitude $y$ (in dB) and phase $\phi$ (in degrees). Four structural conditions are included: a healthy configuration with all masses attached and three progressively damaged configurations created by removing a subset of masses. Across repetitions, the remaining masses are placed at randomly perturbed longitudinal positions along the beam. In our setup, we learn a symbolic regressor for the response magnitude as a function of the excitation frequency and the structural condition. The input variables are the excitation frequency $\omega$ (in kHz) and a scalar damage indicator $d$, representing the percentage of the mass loss for each experiment. The target variable is the magnitude of the inertance response in linear scale.

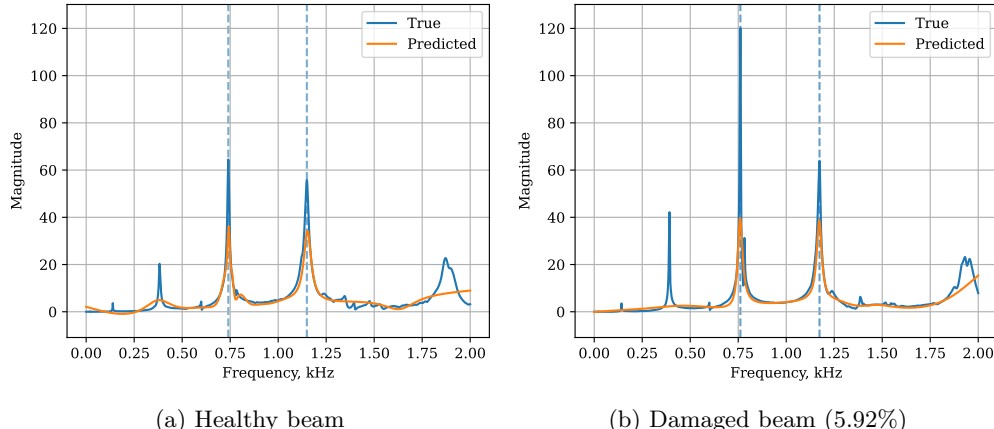

(a) Healthy beam                    (b) Damaged beam (5.92%)

Figure 2: Measured and predicted byt the CEQL model FRF magnitudes for (a) a healthy beam and (b) a beam with 5.92% damage. Dashed vertical lines indicate the locations of two dominant resonance peaks detected from the measured response.

Let $\omega$ denote the excitation frequency (in kHz) and $d$ the damage level expressed as the percentage of removed masses. The training data therefore consist of tuples $(\omega, d, y)$, where $y$ is the linear-scale magnitude of the measured frequency response function.

The model used in this experiment is a symbolic architecture designed to represent the structure of frequency response functions. The predicted response magnitude is modeled as a linear broadband trend combined with a sum of resonant terms

$$\hat{H}(\omega, d) = a\omega + b + \sum_{i=1}^{n} \frac{A_i}{\left(\omega - h_i(d)\right)^2 + \gamma_i},\tag{8}$$

where $a$ and $b$ represent the broadband trend, $A_i$ and $\gamma_i$ are learned coefficients controlling the amplitude and width of the $i$-th resonance, and $h_i(d)$ denotes a symbolic function describing how the corresponding resonance frequency shifts with damage level $d$. The model is initialized with a fixed number of resonant components $n = 20$. During training, periodic pruning of the internal CEQL weights removes connecting edges, effectively reducing the number of active resonant terms. The minimum number of edges is set to 50.

Each function $h_i(d)$ is generated by a CEQL symbolic network that receives the damage variable $d$ and produces a nonlinear expression representing the damage-dependent resonance location. The CEQL component consists of two symbolic layers with a fixed operator library. Each layer uses the operators $\{\mathrm{id}, \mathrm{const}, \mathrm{square}, \mathrm{mul}\}$. Through these layers the model learns compact symbolic expressions for the resonance shifts $h_i(d)$. The resulting architecture therefore balances interpretability and structural simplicity with the expressive capacity required to approximate the measured frequency response.

Figure 2 shows the predicted frequency response functions together with the measured responses for healthy and damaged beam configurations. The symbolic model captures the overall structure of the FRF, including the broadband trend and the dominant resonance peaks. In particular, the model reproduces the two major resonances around 0.75 kHz and 1.18 kHz and correctly captures their shifts across damage levels. The predicted peaks appear smoother and slightly lower in amplitude than the measured responses. This behavior is consistent with the structure of the model in Equation 8, where the damage dependence is introduced only through the resonance location $h_i(d)$, while the numerator coefficients $A_i$ and the denominator constants $\gamma_i$ remain fixed for all damage levels. As a result, the model has sufficient flexibility to adjust the position of each resonance but limited freedom to adapt the exact peak shape or height across damage conditions. Consequently, the symbolic model prioritizes accurate prediction of resonance locations while providing a simplified approximation of the peak amplitudes.

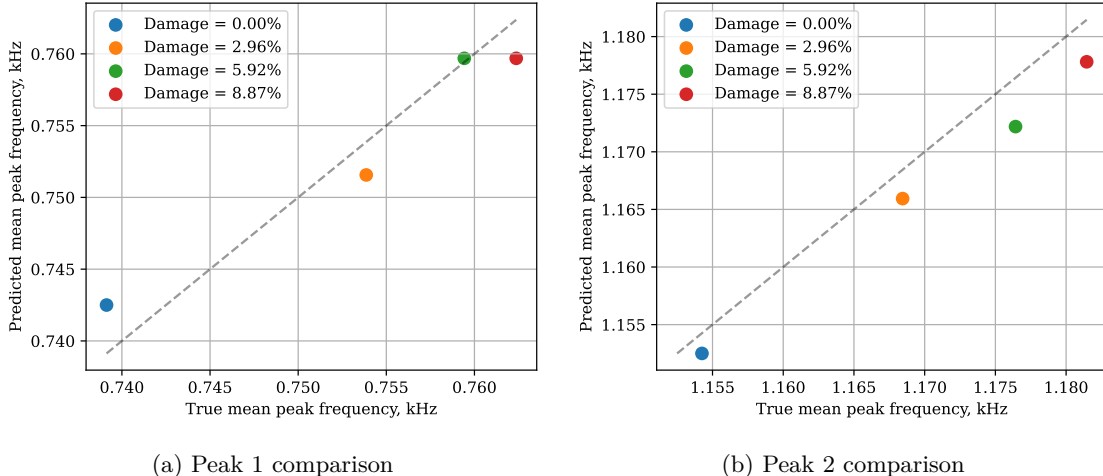

(a) Peak 1 comparison    (b) Peak 2 comparison

Figure 3: Comparison of predicted and measured resonance frequencies for the first (a) and second (b) peaks across all damage levels. Each point corresponds to the mean peak frequency computed from multiple FRF measurements.

The symbolic expression obtained after training and pruning is reported in Appendix D. The resulting model contains several resonant terms whose denominators follow the structure of Equation 8, where the resonance locations are expressed as nonlinear functions of the damage variable $d$. This form allows the model to explicitly describe how modal frequencies shift as the structural condition changes. Several rational terms correspond to resonances located near the dominant peaks observed in the measured frequency response, while the linear terms contribute to approximating the broadband behavior of the FRF. Due to the pruning procedure, the final expression is substantially simpler than the initial architecture, retaining only the resonant components necessary to approximate the measured response while satisfying the minimal number of edges condition.

To quantify how well the symbolic model captures the resonance shifts, Figure 3 compares the predicted and measured mean frequencies of the two main peaks across all damage levels. For each structural condition, the peak frequency is estimated within a fixed window around the expected resonance location and averaged over repeated FRF measurements. The predicted peak locations closely follow the measured values and lie near the identity line, indicating that the model accurately captures the damage-dependent shift of resonance frequencies. Small deviations appear mainly for the healthy configuration and for higher damage levels, but the overall trend across damage states is preserved for both resonances.

## 5 Discussion

The benchmark and ablation results separate several sources of difficulty in symbolic regression. The first is library misspecification. Methods such as SINDy assume that the target expression is sparse in a predefined library of candidate terms. Therefore, their failure on rational, logarithmic, and square-root expressions is largely expected when the corresponding nonlinear structures, including their internal coefficients, are not explicitly present in the library.

The second source of failure is optimization. This is the case for EQL-type architectures, where the relevant operators may be available in the symbolic library, but gradient-based training can still fail to identify the correct structure. For rational expressions, this difficulty appears to stem from real-valued degeneracies around zero-crossings in the denominator, where gradient contributions with opposite signs cancel. The ablation study supports this interpretation: the real-valued variant fails to recover the multivariate rational target, whereas the complex-valued variants recover it in several runs. This suggests that introducing the imaginary dimension changes the optimization geometry in a way that is useful for learning sign-changing denominators.

The ablation study also shows that numerical accuracy and symbolic recovery are not equivalent. Removing the imaginary-weight penalty can still lead to low prediction errors, but eliminates symbolic recovery in the ablation experiment. This indicates that the penalty acts as a structural bias that helps the complex-valued model collapse back toward a real-valued symbolic expression. Similarly, skip connections are not required for representability in the tested rational expression, but they improve stability and compactness by providing shorter paths from the inputs to deeper symbolic layers.

The results further highlight the role of operator-library design. Adding logarithmic operators to a purely rational discovery task reduces the recovery rate and increases the variability of the recovered node count. One possible explanation is that unnecessary operators introduce competing symbolic branches during optimization, so logarithmic and rational terms may partially explain the same variation in the data. This can reduce the apparent importance of required rational terms and make pruning less reliable. We interpret this as an empirical indication that operator-library design and pruning interact strongly in gradient-based symbolic models.

From an application perspective, this behavior is especially relevant in engineering settings, where rational dependencies and poles are often fundamental. Recovering poles explicitly as symbolic denominators improves interpretability and enables direct reasoning about resonance behavior and stability margins.

At the same time, the results expose clear limitations of gradient-based symbolic learning that are not resolved by the proposed extension. Trigonometric operators remain particularly challenging to identify over wide input domains. The oscillatory nature of $\sin(\cdot)$ and $\cos(\cdot)$ leads to frequent sign changes in per-sample gradients, which in turn causes substantial cancellation when gradients are averaged across data. As a consequence, optimization provides weak or misleading signals for frequency and phase parameters unless initialization is already close to the target solution. This observation is consistent with prior reports that successful recovery of trigonometric expressions in EQL-based models often relies on carefully chosen initializations, and it suggests that architectural or optimization-level modifications are required to address this limitation more fundamentally.

A related challenge concerns sparsity enforcement. While smooth sparsity penalties and pruning strategies are effective in reducing expression size, the results indicate that their interaction with optimization remains nontrivial. Aggressive regularization or early pruning can irreversibly remove correct components of an expression, whereas insufficient regularization leaves residual terms that obscure the underlying structure. The ablation results reinforce this point, since both unnecessary operators and insufficient structural bias can reduce symbolic recovery even when numerical errors remain low. More principled pruning and regularization strategies are therefore needed for reproducible recovery of minimal expressions.

## 6 Conclusion

This work introduced Complex Equation Learner (CEQL), an extension of the Equation Learner framework that uses complex weights together with a tailored optimization strategy to expand the class of symbolic operators that can be learned reliably with gradient-based methods. The primary challenge addressed is the failure of real-valued optimization in the presence of sign-changing gradients and near-singular structures, most prominently those arising from division. By optimizing in the complex domain and projecting predictions back onto the real axis, CEQL mitigates gradient-cancellation pathologies that prevent standard EQL variants from discovering rational expressions with sign-changing denominators and singular behavior.

Across synthetic benchmarks with singularity-inducing operators, CEQL trains stably and recovers compact analytical expressions in regimes where real-valued baselines become unreliable. On the benchmark tasks considered, CEQL exhibits low prediction error in both interpolation and extrapolation regimes (Table 2) and high symbolic recovery rates (Table 3), indicating that the learned expressions often capture the underlying functional structure. The ablation study further shows that complex-valued weights are essential for recovering sign-changing rational denominators, while the imaginary-weight penalty, skip connections, and a focused operator library improve the stability of symbolic recovery. In addition, the cantilever-beam frequency response experiment shows that CEQL can recover rational near-pole structure from real measured data, producing interpretable expressions that support direct analysis of damage-dependent resonance shifts.

Several directions for future work remain. Learning periodic trigonometric structures continues to be challenging due to periodic sign changes that lead to weak or canceling gradient signals over large domains, suggesting the need for surrogate operators or continuation-based training schemes to robustly recover frequency and phase parameters. Enforcing sparsity in a stable and reproducible manner also remains difficult; improved pruning criteria and regularization schedules are required to consistently obtain minimal symbolic expressions. More broadly, the complex-parameter perspective provides a general mechanism for reshaping ill-conditioned real-valued loss landscapes and may prove useful for other non-smooth operators beyond symbolic regression. Finally, because CEQL outputs explicit rational forms, it is naturally aligned with rational approximation and pole-matching tasks, where interpretable denominators are primary objects of interest.

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

## A    Illustration of Complex-Domain Optimization

To illustrate the effect of complex-valued optimization for division, we consider approximating $f(x) = 1/x$ with the model $\hat{f}(x) = 1/(x + a)$. Figure 4 compares the trajectory of the parameter $a$ under Adam when $a$ is constrained to $\mathbb{R}$ and when $a$ is allowed to vary in $\mathbb{C}$. The target function is sampled at 100 points uniformly from $[-3, 3]$, and the loss is defined as the $\ell_2$ approximation error.

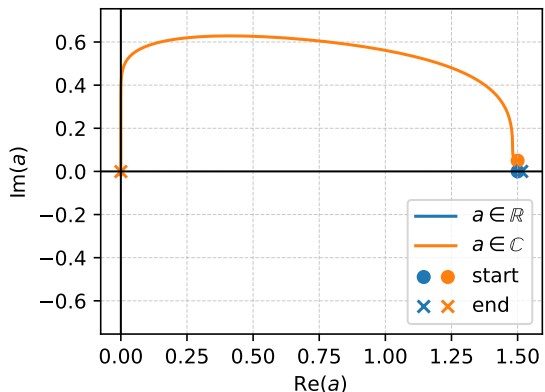

Figure 4: Optimization trajectories of the parameter $a$ in the complex plane for the task of approximating $f(x) = 1/x$ with $\hat{f}(x) = 1/(x + a)$.

## B  Hyperparameters and Experimental Settings

This section reports all hyperparameters used for the symbolic regression benchmarks E-1–E-10. Unless explicitly stated, all hyperparameters are fixed across benchmark expressions and across independent runs.

### B.1  Complex Equation Learner (CEQL)

Table 5: CEQL hyperparameters used for benchmarks E-1–E-10.

| Component | Setting |
|---|---|
| Optimizer | Adam |
| Learning rate (LR) | $10^{-2}$ |
| LR scheduler (phase 3) | ReduceLROnPlateau |
| LR scheduler patience | 2000 |
| LR scheduler factor | 0.1 |
| Minimum LR | $10^{-5}$ |
| Convergence threshold | $10^{-7}$ |
| Loss | Pivoted Relative Mean Squared Error (PR-MSE) |
| Epochs | 100,000 (phase 1), 200,000 (phase 2), 50,000 (phase 3) |
| Im($w$) lambda | $10^{-10}$ (Phase 1), $10^{-3}$ (Phase 2), $10^{3}$ (Phase 3) |
| $\|w\|_1$ lambda | $10^{-10}$ (Phase 1), $10^{-7}$ (Phase 2), $10^{-7}$ (Phase 3) |
| Log, sqrt argument penalty | $10^{-10}$ (Phase 1), $10^{3}$ (Phase 2), $10^{3}$ (Phase 3) |
| Symbolic layers | 2 |
| Library operators (per layer) | $(const \times 2, (.)^2 \times 2, const \times 2, mul \times 2)$, $(id, \log . \times 2, \sqrt{.} \times 2, \div \times 2)$ |
| Pruning enabled | Phase 2 |
| Pruning interval, epochs | 10,000 |
| Pruning fraction | 0.1 |
| Minimal number of edges | 15 |

The benchmark experiments were run on available computational resources suitable for each implementation. Since the methods rely on different software stacks and were not all executed on identical hardware, wall-clock time is not used as a comparative performance metric. In practice, a single run for each method was on the order of ten minutes for a single run. The main quantitative comparison therefore focuses on interpolation error, extrapolation error, node count, and symbolic recovery rate.

### B.2  PySR

Table 6: PySR hyperparameters used for benchmarks E-1–E-10.

| Component | Setting |
|---|---|
| Number of iterations | 1000 |
| Population size | 27 |
| Number of parallel populations | 200 |
| Max sequence size | 20 |
| Unary operators | $(\log, \sqrt{\cdot}, \cdot^2)$ |
| Binary operators | $(+, -, \times, \div)$ |
| Element-wise loss | $(x - y)^2$ |
| Model selection | best |
| Crossover probability | 0.0259 |
| Tournament selection n | 15 |
| Tournament selection p | 0.982 |

## B.3 EQL with division

Table 7: EQL$_\div$ hyperparameters used for benchmarks E-1–E-10.

| Component | Setting |
|---|---|
| Optimizer | SGD |
| Learning rate | $10^{-2}$ |
| Number of epochs | 10,000 |
| Batch size | 64 |
| Number of symbolic layers | 2 |
| Units per base function | 3 |
| Unary operators | $(id)$ |
| Binary operators | $(mul)$ |
| Output node | $\div$ |
| $\|w\|_1$ regularization | $10^{-3}$ |
| $\|w\|_2$ regularization | 0 |
| Regularization start epoch | 500 |
| Regularization end epoch | 9500 |
| Validation interval | 100 epochs |
| Division regularization parameter $k$ | 50 |
| Symbolic pruning threshold | $10^{-2}$ |
| Active-unit threshold | $10^{-2}$ |

## B.4 SINDy

Table 8: SINDy hyperparameters used for benchmarks E-1–E-10.

| Component | Setting |
|---|---|
| Polynomial degree | 2 |
| Include interactions | True |
| Include polynomial bias term | True |
| Unary operators | $(\log, \sqrt{\cdot})$ |
| Binary operators | $(\div)$ |
| Sparse optimizer | STLSQ |
| Sparsity threshold | $10^{-2}$ |
| Ridge regularization $\alpha$ | $10^{-6}$ |
| Maximum optimizer iterations | 1000 |
| Normalize library columns | True |
| Log stabilization $\epsilon$ | $10^{-12}$ |
| Division stabilization $\epsilon$ | $10^{-6}$ |
| Use absolute value in $\sqrt{\cdot}$ | True |
| Maximum feature magnitude (clipping) | $10^{6}$ |
| Coefficient zero tolerance | $10^{-12}$ |
| Number of runs per benchmark | 1 |

## C  Recovered Symbolic Expressions

Table 9: Symbolic expressions recovered by CEQL on benchmarks E-1–E-10. For each benchmark, we report the expression obtained in the run with the lowest training MSE across five independent initializations.

| ID | Target expression | Discovered expression |
|---|---|---|
| E-1 | $1.87\,x_1 + 2.01$ | $1.87x_1 + 2.01$ |
| E-2 | $1.56\,x_1 + 1.59\,x_2 - 2.91$ | $1.56x_1 + 1.59x_2 - 2.91$ |
| E-3 | $2.48\,x_1^2 + 1.92\,x_1 - 0.68$ | $2.48x_1^2 + 1.92x_1 - 0.68$ |
| E-4 | $0.55\,x_1^2 + 2.45\,x_1 x_2 + 2.95\,x_2^2 + 1.65\,x_1 + 0.80\,x_2 + 0.86$ | $0.55x_1^2 + 2.45x_1x_2 + 1.64999x_1 + 2.95x_2^2 + 0.8x_2 + 0.86$ |
| E-5 | $-2.05 \log(1.56\,x_1^2 - 0.55\,x_1 - 2.15)$ | $1.0 \cdot 10^{-5} \left(x_1^2\right)^{0.5} + 1.28808 - 2.05 \log\left(2.92419x_1^2 - 1.03097x_1 - 4.03014\right)$ |
| E-6 | $2.31\,\sqrt{2.52\,x_1^2 - 1.52\,x_1 - 2.24}$ | $1.79564\left(4.17046x_1^2 - 2.51552x_1 - 3.70708\right)^{0.5}$ |
| E-7 | $\dfrac{0.53 - 2.94\,x_1}{2.32\,x_1 + 1.80}$ | $\dfrac{0.79672x_1}{-1.0x_1 - 0.77586} - 0.47051 - \dfrac{0.5935}{-1.0x_1 - 0.77586}$ |
| E-8 | $\dfrac{1.00\,x_1 + 2.48\,x_2 - 1.36}{2.26\,x_1 - 0.91\,x_2 + 1.94}$ | $\dfrac{0.59458x_1}{1.0x_1 - 0.40265x_2 + 0.85841} + \dfrac{1.0361x_2}{1.0x_1 - 0.40265x_2 + 0.85841} - 0.1521 - \dfrac{0.4712}{1.0x_1 - 0.40265x_2 + 0.85841}$ |
| E-9 | $\dfrac{2.84\,x_1^2 + 1.84\,x_1 - 2.33}{-0.66\,x_1^2 + 2.94\,x_1 + 1.35}$ | $0.00017x_1 - \dfrac{3.19994x_1}{0.14582x_1^2 - 0.64944x_1 - 0.29822} - 4.30141 - \dfrac{0.76806}{0.14582x_1^2 - 0.64944x_1 - 0.29822}$ |
| E-10 | $\dfrac{-1.08\,x_1^2 - 2.85\,x_1 - 2.08}{2.56\,x_1^2 + 1.78\,x_1 - 0.74}$ | $-\dfrac{1.17925x_1}{1.4382x_1^2 + 1.0x_1 - 0.41573} - 0.42187 - \dfrac{1.34393}{1.4382x_1^2 + 1.0x_1 - 0.41573}$ |

## D   Recovered symbolic model for the FRF experiment

$$\hat{H}(\omega, d) = 5.21654\omega - 0.3154$$
$$+ \frac{0.08445}{\left(-0.0007d^4 + 0.00217d^3 - 0.00386d + \omega - 0.89\right)^2 - 0.01625}$$
$$+ \frac{0.0577}{\left(0.05572d^2 - 0.25854d + \omega - 1.26425\right)^2 - 0.1399}$$
$$+ \frac{0.47169}{\left(0.03456d^2 - 0.37289d + \omega + 0.17194\right)^2 - 0.36363}$$
$$+ \frac{2.42351}{\left(-0.02541d^2 + 0.061d + \omega - 0.23462\right)^2 - 0.0339}$$
$$- \frac{0.76141}{\left(-0.03146d^2 + 0.08937d + \omega - 0.50138\right)^2 + 0.04658} \tag{9}$$
$$- \frac{0.26376}{\left(-0.20983d + \omega - 0.76732\right)^2 + 0.00551}$$
$$+ \frac{0.48817}{\left(-0.27162d + \omega - 0.20312\right)^2 + 0.31504}$$
$$+ \frac{0.08136}{\left(-0.52067d + \omega - 0.76341\right)^2 - 0.14816}$$
$$+ \frac{0.29455}{\left(\omega - 0.69792\right)^2 + 0.06101},$$

