# OpenReview forum: "Complex Equation Learner: Rational Symbolic Regression with Gradient Descent in Complex Domain"
_TMLR — Decision pending for TMLR_

### Review · Reviewer_fPwS · 2026-04-25

**Summary Of Contributions:**

The paper proposes Complex Equation Learner (CEQL), a gradient-based symbolic regression method. The architecture extends Equation Learner (EQL) by allowing model parameters to take complex values during optimization, which can then be projected back to the real axis to obtain the final expressions. The problem it tries to solve is that existing gradient-based SR cannot handle functions with singularities or domain restriction such as log and division.

Strength: manuscript is easy to follow to understand the problem statement, methodology, and experiments.

Weakness: the benchmark dataset is too small with only ten very simple expressions. The presented results are not strong enough to demonstrate how CEQL is better than other existing methods.

**Audience:**

Yes

**Audience Explanation:**

SR is an active area and the complex parameterization is an interesting idea to address the problem of gradient-based SR.

**Broader Impact Concerns:**

None.

**Claims And Evidence:**

No

**Claims Explanation:**

The benchmark is too small and too simple to support the claim. The ten expressions with mostly trivial polynomials and only one log and a handful of divisions cannot yield conclusive claim that the method can reliably handle functions with singularities. Also CEQL does not seem to be uniformly better than other methods on MSE too, though the expressions are mostly simple. Besides, the point of SR is interpretability, which is used to recover the underlying laws from data, but the paper puts evaluation focus on MSE as the main metric. Given the limitation in the experiments, It is not clear the benefits of CEQL, as there exists a range of genetic programming based algorithms that can handle functions with singularities naturally and high accuracy. the The frequency-response section runs only CEQL, with no baseline comparison to gauge if CEQL is better.

**Requested Changes:**

- Expand the benchmark dataset to include a wider range of expressions in order to support the claim, as a variety of public SR benchmark sets are available which include a lot of expressions with related singularities.
- Include expression-form recovery as one of the primary metrics in the main body, as all benchmark expressions have ground truth.
- In the MSE table, the comparison of CEQL with others is mixed, do you understand if there is systematical reason for this? A larger benchmark would be useful to tell perhaps.
- Run other SR methods on the frequency-response experiment for comparison.

---

> ### Author Response · Authors · 2026-06-19
> **Requested change 1.**
>
> We have not expanded the benchmark with a common public SR benchmark sets, because this would not directly test the contribution of the paper. CEQL is not proposed as a new general-purpose symbolic regression method intended to outperform all existing SR algorithms across broad benchmark suites. It is a targeted extension of EQL-type gradient-based symbolic regression, designed to address a specific optimization failure mode that appears when expressions contain poles.
>
> For this reason, we use a controlled benchmark constructed around the mechanisms studied in the paper. The current set of expressions already covers the main classes of difficulties that CEQL is claimed to address, namely rational expressions with real-domain poles, logarithmic expressions with domain restrictions, and square-root expressions with domain restrictions. The benchmark also includes polynomial expressions as sanity checks. Thus, the evaluation is designed to test whether the proposed complex-valued parameterization improves EQL-type optimization in the specific problematic regimes targeted by the method. Adding many standard polynomial or well-behaved benchmark expressions would mainly evaluate symbolic-regression capabilities inherited from EQL and would provide limited information about the proposed extension.
>
> We also considered common SR benchmark families such as Koza, Nguyen, Keijzer, Korns, and Vladislavleva [1]. While these benchmarks are useful for general SR evaluation, they are not well aligned with the scope of this work. Many of their expressions do not isolate real-domain poles or domain-restricted operators in the training domain, while others rely on operators such as periodic trigonometric functions, which remain a separate limitation of EQL-type architectures and are not the contribution addressed here. Therefore, using such benchmarks would mix the specific question studied in this paper with unrelated limitations of the underlying EQL framework.
>
> At the same time, we agree that the original benchmark set was limited in dimensionality. Rather than adding a broad benchmark suite that would test many aspects outside the scope of the paper, we added a targeted ablation study on a three-variate rational expression with a sign-changing affine denominator. This experiment extends the evaluation to a multivariate denominator structure while remaining focused on the main question of the paper, namely whether CEQL improves EQL-type optimization for rational expressions with sign-changing denominators. The ablation compares the full method against real-valued EQL with the same architecture and pruning, CEQL without the imaginary-weight penalty, CEQL without skip connections, and CEQL with an enlarged operator library.
>
> We have revised the manuscript to make the scope clearer. The claims are now restricted to improving EQL-type gradient-based optimization for rational and domain-constrained symbolic expressions, rather than establishing a general state-of-the-art SR benchmark result across all expression classes.
>
> **Reference**
>
> [1] Makke, N., & Chawla, S. (2024). Interpretable scientific discovery with symbolic regression: a review. *Artificial Intelligence Review*, 57, 2.

---

> ### Author Response · Authors · 2026-06-19
> **Requested change 2.**
>
> Thank you for your recommendation. We have added expression-form recovery to the main body and now report the symbolic recovery rate together with the node count in Table 3. The recovery rate measures whether the recovered expression is algebraically equivalent to the ground-truth expression. To compute it, we implemented an automatic equivalence check between the recovered expression and the target expression. Since standard SymPy simplification and expansion were not sufficient for all cases, especially for rational and logarithmic expressions, we additionally transform the expressions into canonical forms before checking equivalence. This gives a stricter structural metric than numerical error alone and directly evaluates whether the underlying symbolic form was recovered.

---

> ### Author Response · Authors · 2026-06-19
> **Requested change 3.**
>
> Thank you for raising this point. We agree that the comparison based on MSE alone is mixed, and we have revised the manuscript to make this observation more transparent. Specifically, we replaced the previous Table 2 with a more detailed version that now reports interpolation MSE, extrapolation MSE, and their standard deviations over independent runs. We also added Table 3, which reports node count and symbolic recovery rate. Thus, the revised results separate numerical accuracy from expression complexity and exact symbolic recovery.
>
> Importantly, we do not view the mixed comparison with PySR as contradictory to the goals of the paper. CEQL is not proposed as a replacement for GP-based symbolic regression methods, nor do we claim that it should outperform PySR across all expression classes. Rather, CEQL is a targeted extension of EQL-type gradient-based symbolic regression designed to address a specific optimization challenge, namely learning expressions involving division, logarithms, square roots, and other operators that introduce singularities or domain restrictions into the computational graph.
>
> From this perspective, the relevant trend is clearer. Compared with $EQL_{\div}$, CEQL substantially improves recovery on rational and domain-constrained expressions, which are precisely the cases that motivate the complex-valued parameterization. Compared with SINDy, CEQL is more suitable for expressions where the nonlinear structure contains learnable internal parameters, such as denominators, logarithmic arguments, or square-root arguments. SINDy performs well when the correct terms are present in the prescribed library, but it is disadvantaged when these nonlinear structures are absent from the library.
>
> The comparison with PySR is expected to be more nuanced. GP-based methods such as PySR perform discrete structure search and are not affected by the same real-valued gradient pathologies that motivate CEQL. Therefore, there is no reason to expect CEQL to systematically outperform PySR on all benchmark expressions. The contribution of CEQL is instead to extend the applicability of differentiable, gradient-based symbolic regression to expression classes that are difficult for standard EQL-type models.
>
> We have revised the manuscript to make this distinction explicit. The primary contribution is not a universally superior symbolic regression algorithm, but an extension that improves the robustness and expressiveness of EQL-style differentiable symbolic regression in the presence of singular and domain-constrained operators. More broadly, differentiable symbolic models offer capabilities complementary to GP-based approaches, including integration into neural architectures, end-to-end gradient-based training, and the incorporation of differentiable constraints or downstream objectives. We therefore view CEQL as expanding the scope of gradient-based symbolic regression rather than competing directly with GP-based methods on every benchmark.

---

> ### Author Response · Authors · 2026-06-19
> **Requested change 4.**
>
> Thank you for this suggestion. We considered adding this comparison, but decided not to include it because it would not directly evaluate the purpose of the FRF experiment.
>
> The FRF experiment has a different role from the synthetic benchmark. In the benchmark section, the ground-truth symbolic expressions are known, which makes it possible to compare methods in terms of both numerical accuracy and symbolic recovery. This is the appropriate setting for controlled comparisons between CEQL, PySR, SINDy, and $EQL_{\div}$.
>
> In contrast, the measured FRF data do not have a known ground-truth symbolic expression. Therefore, a comparison with other SR methods on this experiment would mainly reduce to comparing predictive errors, such as MSE. Such a comparison would measure curve-fitting performance, but it would not directly assess symbolic recovery, pole identification, or the interpretability of the recovered rational structure.
>
> The purpose of the FRF experiment is instead to demonstrate a realistic use case of CEQL. Specifically, we show that CEQL can be combined with a physically motivated rational structure to recover interpretable resonance terms and track damage-dependent shifts in modal behavior. The focus is therefore on the structure of the recovered model and its consistency with the expected physics of frequency response functions, rather than on benchmark-style error comparison.
>
> For this reason, we keep the controlled method comparison in the synthetic benchmark section, where the true expressions are known and symbolic recovery can be evaluated directly. We have revised the manuscript to clarify this distinction and to state explicitly that the FRF experiment is intended as an application-oriented demonstration of interpretable rational model discovery, not as a general SR benchmark.

---

### Review · Reviewer_zLab · 2026-04-28

**Summary Of Contributions:**

The paper extends Equation Learner (EQL) models to the complex field to handle optimization issues arising from sign-changing structures of gradients of certain functions and functions with domain restrictions. Specifically, the modified CEQL model allows complex values in its parameters and intermediate outputs to bypass the aforementioned optimization difficulties in the real domain, and projects the complex output back to its real part in each modified component. The proposed model is shown to perform better than the EQL (with division) baseline on a few SR benchmark expressions, while maintaining comparable performance with the genetic-programming-based PySR baseline.

**Audience:**

Yes

**Audience Explanation:**

The paper addresses the important problem of equation discovery with symbolic networks. It identifies critical failure modes of the existing EQL model and proposes some reasonable remedies.

**Claims And Evidence:**

Yes

**Claims Explanation:**

The paper claims that performing intermediate calculations in the complex domain can help overcome optimization obstacles and domain restrictions in the vanilla EQL model. This is supported by both clear textual explanations in Sec 2-3 and the experiment results in Sec 4, which show that CEQL clearly outperforms EQL.

**Requested Changes:**

* Sec 2.2: The motivation for lifting the domain restrictions is unclear. The authors mentioned that "the true target expression may change sign". Does this mean that, for example, the target function in some SR task could include something like $\log x$ for $x < 0$, or generically, $x \in \mathbb C$? However, if that is the case, no such examples are included in the experiment benchmark to demonstrate the advantage of CEQL. The authors should include such an example in the experiment. But please let me know if I have misinterpreted this point in Sec 2.2.
* Sec 3.1, paragraph 2: broken cross-ref to appendix.
* Eq (2): explicitly define $\mathfrak R$.
* Table 2: including the standard deviations of the metrics, at least for node complexity, would be helpful to improve the statistical significance of the results.
* A lot of existing work on symbolic regression uses symbolic accuracy as a metric to evaluate their methods, namely, whether the learned equation is mathematically equivalent to the ground truth. If possible, the authors should also report this metric in addition to the numerical errors in Table 2.

---

> ### Author Response · Authors · 2026-06-19
> **Requested change 1.**
>
> Thank you for raising this point. We agree that the motivation in Section 2.2 was not sufficiently clear and may have suggested that CEQL is intended to model complex-valued target functions. This is not the case.
>
> Neither the inputs nor the target expressions considered in this work are complex-valued. All datasets, target values, and final recovered expressions are real-valued. The complex domain is used only internally as an optimization mechanism.
>
> The difficulty comes from the fact that, in EQL-type models, the arguments of operators such as division, logarithm, and square root are learned intermediate expressions. During training, these intermediate expressions change continuously as the weights are updated. Therefore, even when the target expression is real-valued and evaluated only on points where it is well-defined, the current model can temporarily produce intermediate arguments that cross a pole or leave the valid real domain. In a purely real-valued implementation, this leads to undefined forward evaluations, unstable gradients, or optimization barriers that can prevent convergence to the correct expression.
>
> This is especially relevant because the optimizer does not know the correct symbolic form in advance. For example, when learning an expression such as $\log(q(x))$, where $q(x)$ is a polynomial that is positive only on part of the input domain, the model must discover both the structure and the coefficients of $q(x)$. During this process, candidate intermediate arguments may become non-positive for some samples, even if the target data themselves are taken only from the valid real domain. Thus, strict real-domain evaluation introduces artificial obstacles caused by intermediate training states, not by the final target expression.
>
> The motivation for CEQL is to remove these optimization barriers. By evaluating logarithms and square roots through their principal complex branches and then projecting the result back to the real axis, the computational graph remains evaluable during training even when intermediate expressions temporarily leave the valid real domain. The final task remains real-valued, but the optimizer is allowed to move through a larger parameter space.
>
> This situation is already represented in the benchmark. Expression E-5 contains a logarithm whose quadratic argument changes sign over the considered interval, and E-6 contains the analogous case for the square-root operator. In both cases, samples are used only where the corresponding real-valued target is defined. However, the valid domain contains a gap, and the learned intermediate expressions can cross invalid regions during optimization. These examples therefore directly test the domain-restriction issue discussed in Section 2.2.
>
> We have revised Section 2.2 and the description of the surrogate operators to make this distinction explicit. The goal of CEQL is not to recover complex-valued target functions, but to improve the optimization of real-valued symbolic expressions whose operators create poles or real-domain restrictions during training.

---

> ### Author Response · Authors · 2026-06-19
> **Requested change 2.**
>
> Thank you for mentioning! We have corrected the broken cross-reference in Section 3.1.

---

> ### Author Response · Authors · 2026-06-19
> **Requested change 3.**
>
> Thank you for the comment. We have added a definition of $\Re$ at the first paragraph of Section 3.1.

---

> ### Author Response · Authors · 2026-06-19
> **Requested change 4.**
>
> We agree that the variability across runs should be shown in the main body of the manuscript. We have therefore replaced Table 2 with the corresponding table from the appendix. The updated table now reports the mean and standard deviation of the evaluation metrics across five independent runs. Please note that SINDy is deterministic, and we therefore report its results from a single run. Thank you for your suggestion.

---

> ### Author Response · Authors · 2026-06-19
> **Requested change 5.**
>
> Thank you for the comment! We have added the symbolic recovery rate to the manuscript in the new Table 3, together with the node count. We implemented an automatic symbolic comparison procedure. It checks whether the recovered expression is equivalent to the ground-truth expression after coefficient rounding, simplification, expansion, and canonicalization of rational, logarithmic, and square-root expressions. However, automatic symbolic equivalence checking is still fragile. Therefore, we report recovery rate as an additional diagnostic metric, but we do not make strong claims based on this metric.

---

### Review · Reviewer_uFRC · 2026-05-05

**Summary Of Contributions:**

**Summary**
The paper proposes Complex Equation Learner (CEQL), an extension of the Equation Learner framework that uses complex-valued weights to improve gradient-based symbolic regression with problematic operators such as division, logarithm, and square root. The central claim is that complex-domain optimization can bypass real-axis gradient degeneracies, especially for rational expressions with poles. The method is evaluated on 10 synthetic symbolic-regression benchmarks containing polynomials, logarithms, square roots, and rational functions, and on a real frequency-response-function dataset from a cantilever beam. The reported results show strong performance of CEQL on interpolation and extrapolation MSE, especially compared with real-valued EQL with division and SINDy; PySR remains competitive and sometimes stronger on extrapolation.

**Strengths**
- **S1:** The use of complex-valued weights to avoid real-domain optimization pathologies is conceptually appealing. The paper clearly identifies an important limitation of EQL-style symbolic regression: real-valued optimization struggles with division and singular structures because gradient contributions can cancel across samples. The toy analysis for learning (1/x) with (1/(x+a)) gives an intuitive motivation for the complex extension.
- **S2:** Many gradient-based SR methods either avoid division/log/square-root operators or use constrained/safe variants that restrict the hypothesis class. The paper targets this exact weakness and seeks to broaden the operator set while preserving differentiability and symbolic extractability.
- **S3:** On the synthetic benchmark, CEQL obtains very low interpolation and extrapolation errors for rational expressions E-7 through E-10, where SINDy and EQL÷ perform poorly. The recovered symbolic expressions in Appendix D are often close to the target expressions, at least up to scaling/reparameterization.

**Weaknesses**
- **W1:** The synthetic evaluation uses only 10 expressions, mostly low-dimensional, which is insufficient for broad claims about symbolic-regression scalability and robustness.
- **W2:** The paper claims favorable scaling over GP-type methods but does not test scaling with input dimension, expression depth, noise level, or operator-library size.
- **W3:** The comparison omits key metrics such as wall-clock time, function evaluations, success rate, expression equivalence, and hyperparameter sensitivity.
- **W4:** The paper does not isolate the contribution of complex-valued optimization from skip connections, pruning, sparsity penalties, imaginary penalties, and other design choices.
- **W5:** The method relies on complex logarithm and square root, but branch choices, branch cuts, and possible discontinuities during training are not experimentally analyzed.

**Audience:**

Yes

**Audience Explanation:**

Yes. Some TMLR readers interested in symbolic regression, scientific machine learning, and interpretable model discovery would likely find the paper relevant. The idea of using complex-valued optimization to make EQL-style symbolic regression more stable for rational, logarithmic, and square-root expressions is interesting and potentially useful.

**Claims And Evidence:**

Yes

**Claims Explanation:**

The submission provides reasonable evidence for its main claims. The synthetic results show that CEQL handles division, logarithms, and square roots better than standard EQL-style methods in the tested cases. The frequency-response experiment is also a helpful real-world example, since it shows the method recovering an interpretable rational structure. I think the results are promising, but the evidence would be stronger with more benchmarks, clearer scaling studies, and a few targeted ablations. Overall, the claims are generally plausible and supported by the current experiments, though some of the broader claims could be better validated.

**Requested Changes:**

1. Evaluate CEQL on a larger and more diverse symbolic-regression benchmark, including higher-dimensional inputs, deeper expressions, larger operator libraries, different noise levels, and more varied sampling regimes.
2. The paper should empirically support its claim of better scaling than GP-type methods by varying input dimension, expression depth, operator-library size, and training-set size, while reporting accuracy and runtime.
3. For CEQL, PySR, SINDy, and EQL÷, report wall-clock time, number of function evaluations or optimization steps, success rate across runs, expression size, expression equivalence when possible, and hyperparameter sensitivity.
4. The discussion should state more explicitly that SINDy is disadvantaged when the correct nonlinear library terms are absent, so its failure on rational, logarithmic, and square-root examples is largely expected.
5. Include controlled ablations comparing CEQL against real-valued EQL with the same architecture and pruning, CEQL without the imaginary penalty, without skip connections, without iterative pruning, and with complex denominators only versus fully complex weights.
6. The gradient-cancellation argument should be presented as an intuition unless stronger guarantees are proved. The treatment of complex logarithm and square root also needs analysis of branch choices, branch-cut crossings, and how such events are handled during training.

---

> ### Author Response · Authors · 2026-06-19
> **Requested change 1.**
>
> We agree that a larger benchmark covering higher input dimensions, deeper expressions, larger libraries, noise levels, and different sampling regimes would be necessary for broad claims about general SR robustness. However, this is not the objective of the present paper. CEQL is proposed as a targeted extension of EQL-type gradient-based symbolic regression, aimed at improving optimization for expressions with division, poles, logarithms, and square roots.
>
> For this reason, we did not add a broad public benchmark suite. Such benchmarks would mix the specific issue studied in this paper with many other factors, including polynomial recovery, trigonometric recovery, noise robustness, search-space scaling, and general SR model selection. These aspects are important, but they are not the contribution addressed here. The current benchmark set already covers the main cases for which CEQL is proposed, namely rational expressions with real-domain poles, logarithmic expressions with domain restrictions, and square-root expressions with domain restrictions.
>
> At the same time, we have strengthened the evaluation within the intended scope. We added a controlled ablation study on a three-variate rational expression with a sign-changing affine denominator. This extends the original benchmark beyond one- and two-dimensional examples and directly tests multivariate rational recovery. The study also evaluates the effect of a larger operator library by adding logarithmic operators to a purely rational recovery task. The results show that the real-valued EQL variant fails to recover the target, while CEQL recovers it in most runs, and that unnecessary operators can reduce symbolic recovery by enlarging the optimization and pruning problem.
>
> We have revised the manuscript to make this scope explicit. The claims are now restricted to improving EQL-type gradient-based optimization for rational and domain-constrained expressions, rather than establishing CEQL as a general-purpose state-of-the-art SR method across broad benchmark suites, noisy settings, or arbitrary expression classes.

---

> ### Author Response · Authors · 2026-06-19
> **Requested change 2.**
>
> We agree that such a claim would require a dedicated scaling study varying input dimension, expression depth, operator-library size, and training-set size, together with accuracy and runtime measurements. Since this is not the focus of the present paper, we have removed the claim that CEQL scales more favorably than GP-type methods. We have also revised the related wording in the Introduction and removed the corresponding discussion paragraph. The revised manuscript now limits the comparison with PySR, $EQL_{\div}$, and SINDy to the benchmark problems considered in the paper and focuses the contribution on stable gradient-based recovery of rational expressions.

---

> ### Author Response · Authors · 2026-06-19
> **Requested change 3.**
>
> We have expanded the experimental reporting in the revised manuscript. Expression size is reported through node count (NC), while success rate and algebraic equivalence are reported through symbolic recovery rate (RR). Since standard SymPy simplification was not sufficient to canonicalize all recovered expressions, we implemented additional equivalence-checking code for the benchmarked cases.
>
> The optimization budgets and hyperparameters for CEQL, PySR, SINDy, and $EQL_{\div}$ are reported in the appendix, including CEQL epochs, PySR iterations, $EQL_{\div}$ epochs, and SINDy optimizer iterations. We do not report a single common number of function evaluations, since this quantity is not directly comparable across gradient-based training, population-based symbolic search, and fixed-library sparse regression.
>
> We also added an approximate runtime statement. Because the methods use different software stacks and were run on suitable available computational resources, wall-clock time is reported only as an implementation-scale reference. For the considered benchmarks, a single run was on the order of ten minutes across the methods.
>
> We did not add a full hyperparameter-sensitivity study, since that would constitute a separate benchmarking study. Instead, the revised manuscript includes controlled ablations of the CEQL components directly relevant to this work, including complex-valued weights, the imaginary-weight penalty, skip connections, and operator-library choice.

---

> ### Author Response · Authors · 2026-06-19
> **Requested change 4.**
>
> We agree that this point needed to be stated more clearly. We have revised the Results section to clarify that SINDy performs well when the target expression is sparse in the prescribed library, but is disadvantaged when the correct nonlinear features are absent. In particular, its poor performance on the rational, logarithmic, and square-root benchmarks is largely expected, since the required terms, including their internal coefficients, are not learned but would have to be specified in the fixed library in advance. We have also reworked the Discussion to distinguish this library misspecification issue from the optimization failures observed in EQL-type models, and to state more generally that this is a limitation of the linear SR methods outside their prescribed fixed library.

---

> ### Author Response · Authors · 2026-06-19
> **Requested change 5.**
>
> We have added a new ablation study in Section 4.2. The study uses a randomly generated three-variate rational expression with a sign-changing affine denominator, so the experiment directly tests multivariate rational recovery.
>
> The added study compares the full CEQL model with a real-valued EQL variant using the same architecture and pruning procedure, CEQL without the imaginary-weight penalty, CEQL without skip connections, and CEQL with an enlarged operator library including logarithmic terms. We report interpolation MSE, extrapolation MSE, modal extrapolation MSE, node count, and symbolic recovery rate across five independent runs.
>
> The results show that the real-valued variant fails to recover the target expression in all runs, while the full CEQL model recovers it in four out of five runs. This supports the main motivation of the paper that complex-valued optimization helps with sign-changing rational denominators. The variant without the imaginary-weight penalty attains low numerical error but zero symbolic recovery, showing that numerical fitting and symbolic recovery are not equivalent. The variant without skip connections preserves the same recovery rate but increases error and node count, indicating that skip connections improve optimization stability and compactness. The logarithmic-library variant reduces the recovery rate, which supports the practical observation that unnecessary operators can introduce competing symbolic branches during optimization and pruning.
>
> We did not include a separate variant with complex denominators only because it is almost equivalent to the proposed model for rational expressions. In the CEQL surrogate definitions, imaginary components are already disregarded by all standard real-projected operations, while the division surrogate keeps the complex degree of freedom in the denominator. Thus, for the rational-discovery setting considered in the ablation study, the relevant imaginary degrees of freedom already act through denominator terms. We also did not include a variant without pruning. Without pruning, the extracted expressions are extremely large, not human-readable, and computationally expensive to simplify automatically. Such a variant would therefore not provide a meaningful symbolic recovery comparison, since the main output of the method is the pruned symbolic expression. We therefore focused the ablation study on directly comparable variants that isolate the main components of the proposed CEQL formulation.

---

> ### Author Response · Authors · 2026-06-19
> **Requested change 6.**
>
> We have revised the manuscript to make clear that the gradient-cancellation argument is an optimization intuition, not a theoretical convergence guarantee. The updated text now presents the $1/x$ example as an illustrative mechanism showing why real-valued optimization can become ineffective near sign-changing denominators, and why allowing complex-valued parameters can provide a useful additional direction during training.
>
> The newly added ablation study also makes this intuition more explicit. The real-valued EQL variant fails to recover the multivariate rational expression, while the complex-valued variants recover it in several runs. We therefore present the complex-domain formulation as an empirically supported way to alleviate this optimization pathology, rather than as a proved guarantee.
>
> For logarithms and square roots, we use the principal complex branch. During training, branch-cut crossings are not treated by a separate analytic procedure. The corresponding operators are evaluated through their principal complex surrogates. The imaginary penalty is introduced to bias the learned weights and intermediate activations toward the real axis, so that the extracted expressions remain real-valued and interpretable. We have clarified in the manuscript that we do not provide a full theoretical analysis of branch-cut crossings, and that the behavior of these operators is evaluated empirically. We also discuss the ablation without the imaginary penalty, which shows that this regularization is important for stable symbolic recovery.